# A drug-inducible sex-separation technique for insects

Nikolay P. Kandul[1], Junru Liu[1], Alexander D. Hsu [2], Bruce A. Hay[2] & Omar S. Akbari [1,2,3✉]

Here, we describe a drug-inducible genetic system for insect sex-separation that demonstrates proof-of-principle for positive sex selection in *D. melanogaster*. The system exploits the toxicity of commonly used broad-spectrum antibiotics geneticin and puromycin to kill the non-rescued sex. Sex-specific rescue is achieved by inserting sex-specific introns into the coding sequences of antibiotic-resistance genes. When raised on geneticin-supplemented food, the sex-sorter line establishes 100% positive selection for female progeny, while the food supplemented with puromycin positively selects 100% male progeny. Since the described system exploits conserved sex-specific splicing mechanisms and reagents, it has the potential to be adaptable to other insect species of medical and agricultural importance.

[1] Division of Biological Sciences, Section of Cell and Developmental Biology, University of California, San Diego, La Jolla, CA 92092, USA. [2] Division of Biology and Biological Engineering, MC 156-29, California Institute of Technology, Pasadena, CA 91125, USA. [3] Tata Institute for Genetics and Society-UCSD, La Jolla, CA, USA. ✉email: oakbari@ucsd.edu

nsects play an important role in genetic research, as they have short life cycles and are simple to work with and contain. Their experimental tractability and similarity of biological pathways to those in humans make them great model systems for basic research. The *Drosophila melanogaster* fruit fly was introduced by Thomas Morgan for the study of heredity in the early 20th century[1,2]. Since then, a plethora of genetic tools and assays accumulated and shared by the fly community transformed *Drosophila* into one of the most widely used genetic model systems. The high level of gene conservation between flies and humans, in conjunction with a single gene ortholog in flies versus multiple paralogs in humans and mice[3,4], placed the *Drosophila* model at the forefront of studies of conserved gene functions, including research into human development and diseases[3–6]. In addition to serving as a relatively simple translational model organism, *Drosophila* provides a model system for diverse insect pest and disease-vector species and frequently acts as a proof-of-concept system for innovative technologies.

The identification and separation of male and female insects are necessary in any genetic study. In *Drosophila* and many other insect species, special care must be taken to separate female flies before they mate, as females will store the sperm from the first mating in the spermatheca. However, current techniques for the sex sorting of insects, which often require sorting them by hand, are labor intensive, time consuming, and error prone, making this step a limiting factor in insect-based studies. Methods for rapidly sex separating insects could simplify these studies and allow for bigger sample sizes than are currently feasible.

Over the years, a diverse array of sex-based sorting methods have been developed for several insect species. First, mechanical separation makes use of physical differences between the two sexes, which may include morphology (size[7] and shape), coloration[8,9], hatch timing[10], and behavioral differences (female blood feeding[11] and male swarming)[12–15]. Second, various genetic approaches have been developed. In a classic genetic sex separation (GSS) approach, a conditional lethal transgene conferring resistance to the insecticide dieldrin (*Rdl*) was translocated to the Y chromosome through irradiation-induced chromosomal rearrangements in multiple *Anopheles* species. This permitted only male survival when exposed to dieldrin[12,16,17]. Another GSS approach uses transgenic insects harboring fluorescent markers either genetically linked to sex chromosomes[16,18,19] or with sex-specific expression[20] that can be mechanically sorted. Finally, sex separation can also be achieved by negative selection against females using conditional sex-specific lethal transgenes that are repressed by continuous tetracycline feeding (a Tet-Off system), like in a few Tephritid fruit flies[21–24] and the yellow fever mosquito *Aedes aegypti*[25], or activated by heat treatments, such as a *temperature sensitive lethal* (*tsl*) mutation in the Mediterranean fruit fly *Ceratitis capitata* (Medfly)[26–28]. In *Drosophila*, two Y-linked systems were used to remove males and generate virgins females: an inducible lethal (P{hs-hid}Y[29]) and a Gal4-UAS transactivating lethal (P{UAS-rpr.Y}[30]). The sex specificity of the marker expression and conditional lethal transgenes was generated serendipitously by linking it to a sex chromosome[25–27,29–31], or by incorporation of a sex-specific promoter[25], or a female-specific intron into a coding sequence of a lethal transgene[21–24].

Notwithstanding, the existing methods for the sex sorting of insects have many shortcomings and are not easily transferable to other species. The methods based on mechanical separation of different sexes are entirely species specific and cannot be adopted to others. In addition, they are not practical in most insects, since they require optimal rearing conditions, multiple transgenes, have a significant error rate, and are not suitable for high-throughput insect production[13,15,27,30,32], with an exception of the Medfly[28]. Classic GSS approaches were achieved serendipitously in one

species at a time, and thus are species specific. Fluorescent sex sorting requires that each larva is examined and sorted individually, and as a result, this approach is generally not suitable for large-scale programs demanding a high-throughput production. Furthermore, induced chromosomal rearrangements and sex chromosomal linkages are genetically unstable and frequently break down as a result of chromosomal recombination when large numbers of insects are raised[15,27,33]. In the Tet-Off approach, tetracycline must be continually supplied to prevent lethality during mass rearing, which can ablate the microbiota and generate unwanted fitness effects, such as negative impacts on mitochondrial function[34–36]. Therefore, to improve the efficiency of current methods for insect studies, approaches for sex sorting that are genetically stable, suitable for a large-scale insect production, and that can be adopted to different insect species are required[13,15,32].

Here we describe a positive, drug-inducible GSS system for insects and demonstrate its proof-of-principle in *Drosophila melanogaster*. Two genes conferring resistance to specific drugs are expressed in opposite sexes by incorporating sex-specific introns disrupting the coding sequences of drug-resistance genes. In the absence of sex selection, the transgenic strain harboring a sex-sorting gene cassette is maintained on normal food. When insects of a particular sex are desired, the transgenic strain is raised on food supplemented with the corresponding drug. Members of the sex selected against will not be resistant to the selecting drug, resulting in the emergence of adults of the desired sex. The described GSS system will still be susceptible to genetic recombination, chromosomal rearrangements and other loss-of-function mutations, however these types of mutations would be selected against by positive drug selection making this system more robust than previously described GSS approaches.

## Results

**Antibiotics inhibit *Drosophila* development**. To engineer a drug-inducible sex-selection system in *D. melanogaster*, we used two common antibiotic-resistance genes, *puromycin N-acetyltransferase* (*PuroR*) and *aminoglycoside phosphotransferase* (*NeoR*). These genes were previously demonstrated to confer resistance in eukaryotic cells, *D. melanogaster* S2 cells, and *D. melanogaster* larvae to the corresponding water soluble antibiotics, puromycin and geneticin (Fig. 1a), respectively[37–40]. To determine the toxic doses for these drugs in *D. melanogaster*, we first raised wildtype (wt) fly larvae on food supplemented with increasing concentrations of either puromycin or geneticin (0, 0.2, 0.4 mg/ml). From this experiment, we determined that a concentration of 0.2 mg/ml of either drug was toxic, though it permitted the survival of some *D. melanogaster* larvae to adulthood ($2.2 \pm 1.5\%$ for puromycin; and $14.0 \pm 6.5\%$ for geneticin, Supplementary Data 1), while concentrations of 0.4 mg/ml and above completely inhibited development, with almost no larvae able to mature past the first instar stage, and 100% of larvae perishing before adulthood on the supplemented food (for each treatment, expected $n > 500$ found 0; replicates $[N] = 5$; $P < 0.001$; two-sample Student's $t$ test with equal variance: Fig. 1b, Supplementary Data 1).

**PuroR or NeoR rescues the induced lethality**. After determining the toxic doses, we then tested if we could rescue this toxicity by the transgenic expression of antibiotic-resistance genes *PuroR* or *NeoR* integrated into the *D. melanogaster* genome. We engineered a *piggyBac* (PB) transposable element that encoded a constitutive baculovirus promoter *Hr5IE1* that drove expression of dsRed as a selectable marker (*Hr5IE1-dsRED*)[41]. To provide a continuous and ample supply of an antibiotic-resistant protein, we used a

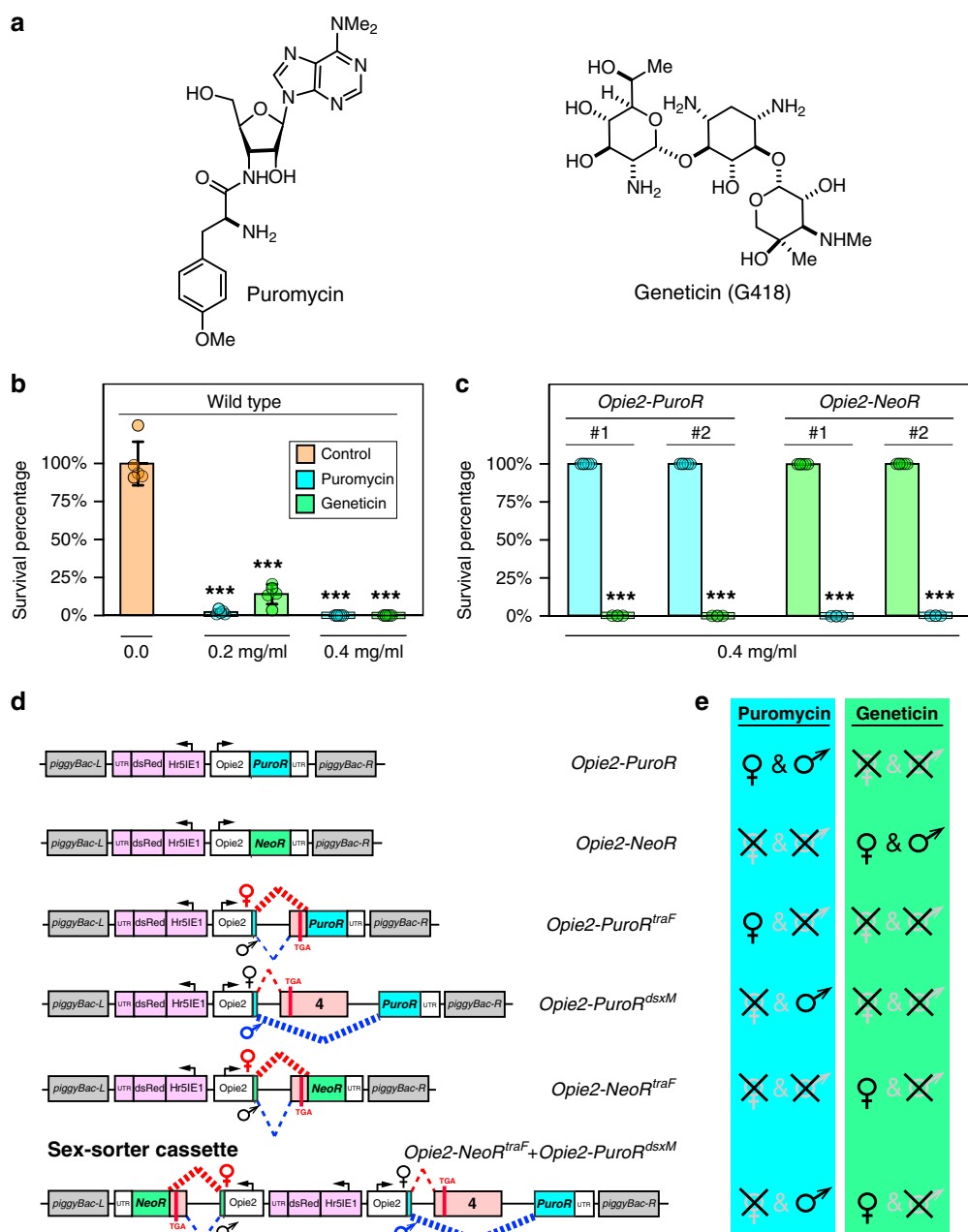

**Fig. 1 Development of sex-sorter cassette in *Drosophila*. a** Chemical structures of puromycin and geneticin (G418). **b** Supplementing fly food with puromycin or geneticin to a final concentration of 0.4 mg/ml completely arrests development of wildtype (wt) *D. melanogaster*. Both drugs are also toxic to wt larvae at 0.2 mg/ml, but a few adult flies do emerge. **c** Fly survival in two independent transgenic lines harboring one copy of either *Opie2-PuroR* or *Opie2-NeoR* mixed with wt flies on food supplemented with 0.4 mg/mL of puromycin or geneticin. The *PuroR* and *NeoR* genes expressed under the *Opie2* promoter rescued transgenic flies harboring one copy of a transgene on the corresponding drug, while all wt flies perished. Bar plots show the average ± one standard deviation (s.d.) over five biological replicates. Statistical significance was estimated using a *t*-test with equal variance. (***$P < 0.001$). **d** Schematic of genetic constructs engineered and tested in the study. The expression of antibiotic-resistance genes (*PuroR* and *NeoR*) throughout *Drosophila* development confers resistance to puromycin and geneticin, respectively, supplemented on fly food. To ensure that functional antibiotic-resistance proteins will be produced only in one or the other sex, sex-specific introns from two sex-determination genes (*tra* and *dsx*) were inserted into coding sequences of *PuroR* and *NeoR*. The entire sequences of female-specific *traF* and male-specific *dsxM* introns (highlighted in pink) are spliced out in the corresponding sex, but some sequences carrying a stop codon (TGA) are retained in the opposite sex (Supplementary Fig. 1). The transgenic flies harboring one copy of a genetic construct were identified by the strong ubiquitous expression of *dsRed* (highlighted in purple). **e** Survival of females and/or males carrying the respective constructs when supplemented with the indicated antibiotic. Source data available in Supplementary Data 1–3.

strong ubiquitous baculovirus promoter *Opie2*[42] to express the *PuroR* or *NeoR* gene (*Opie2-PuroR* or *Opia2-NeoR*) and inserted the gene cassette in an opposite orientation relative to the the marker (*Hr5IE1-dsRED*) to avoid any transcriptional read-through effects. We generated several transgenic lines harboring

a copy of either *Opie2-PuroR* or *Opie2-NeoR* and permitted them to lay eggs on fly food supplemented with either puromycin or geneticin, respectively. When non-balanced transgenic fly lines, which contained both transgenic and wt flies, were raised on food supplemented with either puromycin (0.4 mg/ml) or geneticin

(0.4 mg/ml), only transgenic flies carrying a copy of the resistance gene matched to a supplemented drug emerged, while both wt and non-matched transgenic larvae perished (for each treatment, $n > 500$; $N > 6$; $P < 0.001$: two-sample Student's $t$ test with equal variance; Fig. 1c–e; Supplementary Data 1). Taken together, these results strongly indicated that baculovirus promoter-driven expression of *PuroR* and *NeoR* genes was able to dominantly rescue the larval lethality caused by the consumption of toxic doses of either puromycin or geneticin (Fig. 1).

**Engineering sex-specific expression of *PuroR* or *NeoR*.** We next determined whether we could exploit the endogenous sex-determination machinery to promote antibiotic selection in a sex-specific manner. In *D. melanogaster*, the female-specific intron between *transformer* (*tra*) exons 1 and 2 (*traF*) is spliced out in females. However, in males, some remaining sequence produces a stop codon that prematurely terminates the tra protein[43] (Supplementary Fig. 1). Inversely, the male-specific intron between *double sex* (*dsx*) exons 3 and 5 (*dsxM*) is spliced out in males. However, in females, only a small part is spliced out, leaving the entirety of exon 4, which carries a premature stop codon[44] (Supplementary Fig. 1). Given these known intronic splicing patterns, we hypothesized that if we inserted these sex-specific introns into the coding sequences of *PuroR* or *NeoR*, functional proteins would only be produced in one or the other sex, leading to its survival when exposed to the corresponding drug, while the opposite sex would perish.

To test this hypothesis, we generated plasmids encoding the previously characterized sex-specific introns *traF*[43] or *dsxM*[44] within the coding sequence of *PuroR*, generating two types of synthetic transgenes that would theoretically result in the survival of only female or only male flies when exposed to puromycin, termed *Opie2-PuroR^traF^* and *Opie2-PuroR^dsxM^*, respectively (Fig. 1d, e). We then engineered flies harboring one copy of either *Opie2-PuroR^traF^* or *Opie2-PuroR^dsxM^* and permitted them to develop on fly food supplemented with puromycin. Because the transgene integration site can affect gene expression and sex-specific splicing[45], several transgenic lines were assessed for each construct. Raising two independent lines heterozygous for *Opie2-PuroR^traF^* on food supplemented with puromycin at 0.4 mg/ml resulted in the emergence of only female flies ($n = 421$, $N = 10$, $P < 0.001$: two-sample Student's $t$ test with equal variance; Figs. 1d, e, 2a). Moreover, three *Opie2-PuroR^dsxM^* lines raised on puromycin supplemented food (0.4 and 1.0 mg/ml) each gave rise to significantly male-biased progeny ($n = 741$, $N = 13$, $P < 0.001$: two-sample Student's $t$ test with equal variance; Fig. 2a), and one line produced exclusively male progeny on the higher puromycin concentration, 1.0 mg/ml ($n = 81$, $N = 3$, $P < 0.001$: two-sample Student's $t$ test with equal variance; Fig. 2a; Supplementary Data 1). Notably, when we grew the mixture of best performing *Opie2-PuroR^traF^* or *Opie2-PuroR^dsxM^* transgenic lines (lines #1 and #2, respectively; Fig. 2a) and wt flies on food supplemented with 1.0 mg/ml of puromycin, only female or male transgenic flies marked with dsRed fluorescence emerged, respectively, while no wt larvae survived to adulthood (Fig. 2b).

To determine the versatility of the system and to establish a geneticin-mediated sex-selection system, we inserted the *traF* intron into the coding sequence of *NeoR* and generated transgenic flies harboring *Opie2-NeoR^traF^*. Three independent heterozygous *Opie2-NeoR^traF^* lines raised on food supplemented with geneticin at 0.4 mg/ml resulted in female-biased progeny ($n = 620$, $N = 9$, $P \leq 0.05$: two-sample Student's $t$ test with equal variance; Fig. 2a), and one line enforced 100% female selection at a concentration of 1.0 mg/ml ($n = 188$, $N = 3$, $P < 0.001$: two-sample Student's $t$ test with equal variance; Figs. 1d, e, 2a; Supplementary Data 1). Taken together, these results strongly indicate that by inserting either the *traF* or *dsxM* sex-specific introns into the coding sequence of *PuroR* or *NeoR* the production of functional antibiotic-resistance genes can be sex limited and mediate drug-inducible sex selection.

**A sex-sorter cassette enables selection of both sexes.** To achieve positive drug selection of either sex from a single construct (herein termed a sex-sorter cassette), we next tested whether we could combine the two separate sex-selection systems (*Opie2-NeoR^traF^* + *Opie2-PuroR^dsxM^*; Fig. 1d). We engineered a sex-sorter plasmid, generated three independent transgenic lines harboring the cassette, and tested them by raising heterozygous flies on food supplemented with either puromycin or geneticin at 0.4 and 1.0 mg/ml. For two of three tested lines, only female flies emerged on food supplemented with 1.0 mg/ml of geneticin ($n = 335$, $N = 6$, $P < 0.001$, two-sample Student's $t$ test with equal variance), and only males were recovered from vials containing 1.0 mg/ml of puromycin ($n = 210$, $N = 6$, $P < 0.001$: two-sample Student's $t$ test with equal variance; Supplementary Data 1). The lower drug concentration was not sufficient to enforce the emergence of 100% single-sex progeny for each of three tested transgenic lines harboring only a single copy of the sex-sorter cassette (Fig. 2c). These results strongly indicate that by raising flies harboring one copy of the sex-sorter cassette on the food supplemented with either drug at 1.0 mg/ml, we can dominantly control which sex survives to adulthood.

We next explored an opportunity to lower the drug concentration and still enforce 100% efficient sex sorting by doubling the copy number of the sex-sorting cassette. The homozygous sex-sorter line established from the flies carrying the *Opie2-NeoR^traF^* + *Opie2-PuroR^dsxM^* cassette integrated on the third chromosome (line #3 on Fig. 2c) did not produce any obvious fitness defect and was therefore used for further analysis. We raised the homozygous flies on food supplemented with either puromycin or geneticin, titrating concentrations down from 1.2 mg/ml, and we quantified the percentages of each sex in the emerging flies. We discovered that the addition of puromycin at the final concentrations of 1.2, 1.0, 0.8, 0.6, and 0.4 mg/ml resulted in 100% male progeny. Even 0.2 mg/ml of puromycin caused a significant increase in the male to female ratio, from $43.0 \pm 1.3\%$ to $62.7 \pm 2.15\%$ ($N = 3$, $P \leq 0.001$: two-sample Student's $t$ test with equal variance; Fig. 3a). Inversely, supplementing the food with geneticin to the final concentrations of 1.2, 1.0, 0.8, 0.6, 0.4, and 0.2 mg/ml resulted in 100% female progeny. For geneticin, even 0.1 mg/ml led to a significant increase in female progeny, from $56.1 \pm 1.1\%$ to $78.0 \pm 4.6\%$ ($N = 3$, $P \leq 0.01$: two-sample Student's $t$ test with equal variance; Fig. 3b; Supplementary Data 2).

We investigated why the selection was not effective at lower drug concentrations, and came up with two possible reasons. First, the concentration may be too low to enforce effective selection, as wt flies can also survive at these lower concentrations. We previously found that wt flies cannot be raised on 0.4 mg/ml of either puromycin or geneticin—fly development arrests at the 1st instar larval stage. To test whether wt flies can survive on concentrations lower than <0.4 mg/mL, we raised wt flies on 0.200, 0.100, 0.050, and 0.025 mg/ml of puromycin or geneticin. Notably, while the development of flies raised on food supplemented with each antibiotic was delayed, some flies repeatedly emerged on concentrations ≤0.2 mg/ml for each drug, indicating that these concentrations are indeed too low for complete selection. The second reason the selection might not be effective at lower drug concentrations is that some antibiotics may degrade over time, becoming ineffective in selecting against the opposite sex. In our experiments, we found that puromycin was indeed unstable over time, as we observed that after collecting

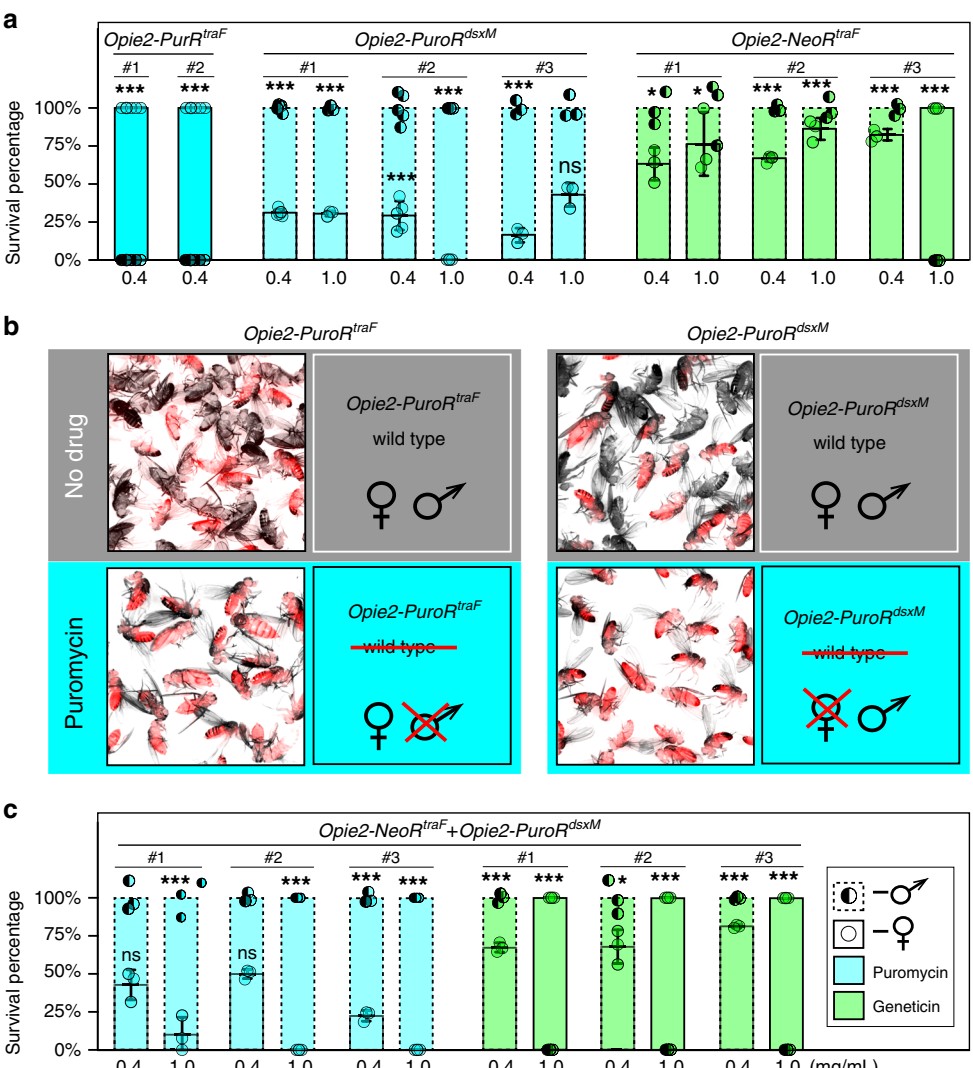

**Fig. 2 Positive selection of a specific sex.** Sex-specific drug resistance is achieved by inserting the female-specific *traF* or male-specific *dsxM* intron into the coding sequences of *PuroR* and *NeoR*. **a** The efficiency of drug-induced sex sorting was assessed for a few independent transgenic lines of the same genetic construct. Transgenic flies harboring one copy of the antibiotic-resistance genes were raised on drug-supplemented food. When sex sorting was not 100% efficient, the higher drug concetration of 1.0 mg/ml was used. **b** Expression of *Opie2-PuroR*$^{dsxM}$ or *Opie2-PuroR*$^{traF}$ transgene rescues only transgenic males or females (red fluorescence) raised on the food supplemented with puromycin, while all wildtype (wt) flies (no red fluorescence) and the transgenic flies of the selected-out sex die during early development. **c** Both antibiotic-resistance genes expressed in the two sexes were combined into one sex-sorter cassette, *Opie2-NeoR*$^{traF}$ + *Opie2-PuroR*$^{dsxM}$. Three independent transgenic lines harboring one copy of the sex-sorter cassette were tested on food supplemented with either puromycin or geneticin at 0.4 and 1.0 mg/ml. We found two transgenic lines that can produce 100% males or 100% females when raised on food supplemented with 1.0 mg/ml of geneticin or puromycin, respectively. Bar plots show the average ± one standard deviation (s.d.) over at least three biological replicates. Statistical significance was estimated using a *t* test with equal variance. ($^{ns}P \geq 0.05$ , *$P < 0.05$, **$P < 0.01$, and ***$P < 0.001$). Source data available in Supplementary Data 1–3.

exclusively male flies from the vials with 0.4 mg/ml of puromycin for eight straight days at +25 °C, a few female flies would emerge starting at day 9 (Supplementary Data 3). However, for geneticin at the same concentration (0.4 mg/ml), only female flies emerged from the vials supplemented with the antibiotic. This indicates that both wt fly survival and antibiotic degradation contribute to the lack of effective sex selection at concentrations lower than 0.4 mg/ml.

**The sex-sorter cassette is not costly to *Drosophila* fitness.** As the fertility of flies is very important for genetic experiments, we tested whether the antibiotic-mediated selection would affect the fertility of the recovered flies. To do so, we repeatedly tested

the fertility of males and females carrying two copies of the sex-sorter cassette raised on the highest tested concentration of puromycin or geneticin (1.2 mg/ml), respectively, by housing them with wt virgin flies of the opposite sex on non-supplemented food. In each case, numerous progeny containing both sexes without any obvious phenotypic defects were obtained. Taken together, these results show that flies of a specific sex can be generated with 100% efficiency by simply raising flies containing two copies of the sex-sorter cassette on food supplemented with either puromycin (male selection) or geneticin (female selection) at 0.4 mg/ml.

To estimate the fitness costs to the carriers of two copies of the sex-sorter cassette, we compared the fitness of homozygous sex-sorter flies to that of wt flies. We found that the survival rates,

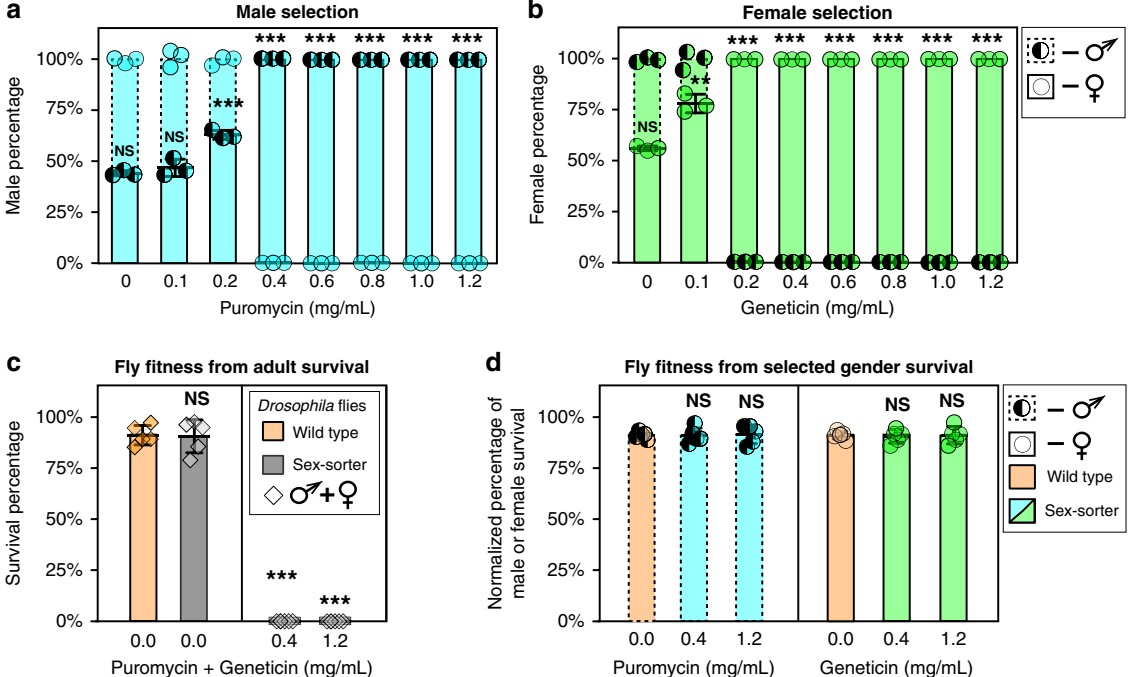

**Fig. 3 Sex selection and fitness of flies carrying two copies of sex-sorter cassette.** The sex-sorter cassette includes $NeoR^{traF}$ and $PuroR^{dsxM}$ genes that confer resistance to the antibiotics puromycin and geneticin, respectively, in a sex-specific manner (Fig. 2a). The $PuroR^{dsxM}$ gene is properly spliced, which results in the expression of the functional PuroR protein only in males, while $NeoR^{traF}$ expresses the functional NeoR protein only in females. To estimate the lowest concentration of an antibiotic at which male or female selection is enforced at 100%, the homozygous sex-sorter flies were raised on various concentrations of puromycin or geneticin. **a** Only male flies emerged from the food supplemented with 0.4 mg/mL or more of puromycin. **b** Raising the same flies on food containing 0.2 mg/ml or more of geneticin resulted in the emergence of only female flies. To compare the fitness of homozygous sex-sorter flies to that of wildtype (wt) flies, the embryo-to-adult survival of both fly types were compared under normal and selective conditions. **c** Embryos of both sex-sorter (gray bars) and wt flies (white bars) survived to the adulthood equally well on food without any antibiotics and died on the food supplemented with both puromycin and geneticin at concentrations of 0.4 and 1.2 mg/ml. **d** The survival of male or female sex-sorter flies under selection treatments was statistically identical to that of the corresponding sex from wt flies raised under normal conditions. Bar plots show the average ± one standard deviation (s.d.) over at least three biological replicates. Statistical significance was estimated using a $t$ test with equal variance. ($^{ns}P \geq 0.05$, $^{*}P < 0.05$, $^{**}P < 0.01$, and $^{***}P < 0.001$). Source data available in Supplementary Data 1–3.

calculated as a percentage of embryos surviving to adults, for both lines raised on non-supplemented food, were not statistically different from wt flies: $91.2 \pm 2.8\%$ of wt embryos survived to adulthood versus $90.4 \pm 8.1\%$ of transgenic sex-sorter embryos ($N = 5$, $P = 0.85$: two-sample Student's $t$ test with equal variance, Fig. 3c). Moreover, no flies survived to adulthood when supplementing fly food with both puromycin and geneticin at either 0.4 or 1.2 mg/ml, which was expected since neither wt nor flies harboring two sex-sorter cassettes are resistant to both drugs simultaneously (Fig. 3c). To further measure fitness, we compared the survival rates of each sex, normalized as a percentage of male or female embryos surviving to adults, compared between wt flies raised on food without any antibiotic and the homozygous sex-sorter flies raised on food supplemented with either puromycin or geneticin. The percentage of wt males that emerged on non-supplemented food was similar to that of sex-sorter males that emerged from the food supplemented with puromycin at concentrations of either 0.4 or 1.2 mg/ml: $91.2 \pm 1.9\%$ versus $90.6 \pm 3.2\%$ ($N = 5$, $P = 0.93$, two-sample Student's $t$ test with equal variance) and $91.2 \pm 4.3\%$ ($N = 5$, $P = 0.77$, two-sample Student's $t$ test with equal variance, Fig. 3d). Similarly, the percentage of hatched wt females was not significantly different from sex-sorter females that emerged on the food supplemented with geneticin at concentrations of either 0.4 or 1.2 mg/ml: $91.4 \pm 1.7\%$ vs $91.2 \pm 2.8\%$ ($N = 5$, $P = 0.93$, two-sample Student's $t$ test with equal variance) and $90.7 \pm 4.5\%$ ($N = 5$, $P = 0.77$, two-sample Student's $t$ test with equal variance; Fig. 3d; Supplementary Data 3).

## Discussion

We describe the proof-of-concept of a GSS approach in Drosophila. Its design is based on the sex-specific expression of two antibiotic-resistance genes ($NeoR$[37] or $PuroR$[38]), which is accomplished by incorporating introns that splice in a sex-specific manner, traF and dsxM. Females or males are positively selected by rescuing their development on food supplemented with either geneticin or puromycin, respectively. The described drug-inducible GSS approach has several advantages over other traditional genetic and mechanical methods for insect sex sorting: genetic stability, positive selection, potential portability across different insect species, low maintenance requirements (i.e. does not need to be supplemented during maintenance), low fitness costs, and potential for adaptability for high-throughput sex sorting.

While this system is still susceptible to mutations such as chromosomal translocations, recombination, and loss-of-function mutations, these types of mutations would likely be selected against when exposed to the antibiotics, as opposed to other GSS which could be broken by these kinds of mutations. For example, traditionally, the construction of genetic sexing lines was based on linking a selectable marker gene, such as insecticide resistance[12,16,17], the tsl[27], and phenotypic loci[27] to a Y-autosome or X-autosome via induced chromosomal translocations. Such engineered lines are not stable, and chromosomal rearrangements will break the genetic linkage between a selective marker and a sex chromosome[13,15,17,27,32,46]. In fact, it was found that when large

numbers of insects were produced from an engineered GSS line, rare chromosomal rearrangements persisted or were selected for and would contaminate the original line, which made it unusable for sex sorting[15,27,28,32]. Given that our GSS mechanistically relies on sex-specific alternative splicing—a translocation of the system to a sex chromosome can change the genetic inheritance pattern, however functional splicing of the sex-specific intron will still be required providing an additional safeguard against breakage. Moreover, any breakage event that causes a loss-of-function mutation in the *NeoR*[37] or *PuroR*[38] antibiotic-resistance genes will be selected out during drug exposure, since the flies harboring loss-of-function mutations will not survive on food supplemented with the corresponding antibiotic assuring that surviving individuals harbor a functional sex-sorter gene cassette. Only gain-of-function mutations, which are very rare, conferring the dominant splicing in a wrong sex may "break" function of the sex-sorter cassette.

The sex-sorter gene cassette was designed to be widely transferable through the use of the PB transposable element, which has been shown to be portable across many insect species[47]. Puromycin and geneticin are general antibiotics that arrest growth in both prokaryotic and eukaryotic cells. The resistance genes for these antibiotics are well established and are routinely used as selection markers for cell-culture transgenesis; they are therefore expected to be effective and portable across a wide variety of insect species[37–40]. Likewise, the *Opie2* and *Hr5IE1* regulatory sequences that direct expression of the sex-specific antibiotic-resistance genes and the transformation marker, respectively, originate from a baculovirus known to infect a large variety of insects[41,42]. Most importantly, the underlying mechanism of sex-specific antibiotic expression via sex-specific alternative splicing is conserved across diverse insect species. For example, both the male-specific *dsx* and female-specific *tra* introns are conserved across many insects[48,49], indicating that it should be possible to transfer this positive sex-selection system into other species. The female-specific *tra* intron of the *Ceratitis capitata* Medfly (*CctraF*) will be possible to transfer into other insect species, since it is functionally conserved[50,51] and does not require the *Drosophila*-specific *sxl* for its splicing[52–54]. In fact, the *CctraF* intron was already used to confer the female-specific transgene splicing to remove females by a negative selection in other fly species[21,22,24,51].

Unlike Tet-Off systems with conditional lethal transgenes[21–25], no antibiotic is required for survival and maintenance of the sex-sorter strain, meaning continuous drug feeding is not necessary during mass rearing[21,25]. The antibiotics, puromycin or geneticin, are supplied only to enforce the sex selection by rescuing the selected sex (positive selection) and "killing" the opposite sex. We demonstrate that drug selection occurs early on, at the first instar stage, and thereafter the surviving sex can be maintained on a regular food. This transient exposure of antibiotics could further reduce any potential fitness costs and could also reduce costs associated with drug selection, as smaller quantities could be used at only the early instar stages—a factor that may play a significant role in large-scale projects.

The sex-sorting gene cassette does not directly affect the fitness of its carriers. Unlike GSS methods that use negative selection[22–25], the sex-sorter cassette does not include a toxin or suicide gene that could leak and affect the organismal fitness. However, the location of transgene integrations in the *Drosophila* genome can affect fitness of transgenic flies[45], and we therefore assessed multiple integration sites for each genetic construct (Fig. 2c). The homozygous sex-sorter transgenic line generated in the study harbors two copies of the sex-sorter cassette (i.e. homozyogous) and has the same egg-to-adult survival rate for each sex as compare with wt flies, even when the transgenic flies were raised on food

supplemented with antibiotics to enforce sex selection (Fig. 3d). We also confirmed that males sex sorted on the highest antibiotic concentration were able to court and mate with wt females. The fitness of sex-sorted males is of great importance[55], since many insect control methods, such as sterile insect technique[55], release of insects carrying a dominant lethal[25], and *Wolbachia*-mediated incompatible insect technique[56–58] rely on male releases. Taken together, our synthetic genetic circuit that relies on the positive, instead of negative, selection can ameliorate some of the effects of a negative selection on organismal fitness.

The positive drug-inducible GSS system presented here complies with the seven key requirements for efficient sex separation technology proposed to Papathanos et al.[13], referred as the "7 Ses". (1) Small: the sex selected against dies early in development and does not compete for resources with the selected sex. (2) Simple: a required sex is positively selected by simply raising the transgenic insects on food supplemented with a drug. (3) Switchable: the sex-sorter transgenic flies can be maintained on a regular food. (4) Stable: the constitutive expression of functional antibiotic-resistance genes in a sex-specific manner guarantees survival of a specific sex on drug-supplemented food, and any loss-of-function mutations in the antibiotic-resistance gene are selected against. (5) Stringent: our data demonstrate that the sex sorting is enforced at 100%. (6) Sexy: the sex sorting happens genetically during insect development and does not required insect handling. It is mediated by positive selection, and its genetic circuit does not include any toxin or suicide genes, which results in minimal to no observable reduction to fitness. (7) Sellable: we designed this system to utilize mechanisms (e.g. mechanistically rely on sex-specific alternative splicing which is conserved in many insects) and components (*piggybac*, a transposon shown to function in many insects), and baculovirus promoters to drive expression of marker and resistance genes, antibiotics (e.g. puromycin and geneticin), and antibiotic resistance genes (e.g. PuroR and NeoR) that should be portable to many insects in the future.

Finally, it has not escaped our attention that the technology proposed herein, with its use of antibiotic resistance, could pose concerns due to the increasing antibiotic resistance worldwide[59,60]. This feature needs to be taken into consideration if this technology is used to generate insects for pest control purposes. Notwithstanding, to mitigate these concerns we have included safeguards in our system to prevent function in prokaryotes. For example, the insertion of an intron into the coding sequence of an antibiotic-resistance gene will block its translation in prokaryotes since the introns will not be spliced. The antibiotic-resistance genes harboring introns in their coding sequences will not confer selective advantage and will not spread through the horizontal gene transfer to and between prokaryotes. Moreover, an antibiotic-resistance gene will have a real advantage only to the extent that the antibiotic concentrations are so high that they cause the death or slow the growth of other species in the environment[60], which would already suggest a problem of a different magnitude—that the antibiotic concentrations in the environment are literally sterilizing the environment, which seems unlikely[61]. We suggest that the insertion of introns into transgenes to break their coding sequences could be both a useful strategy to improve their expression levels in *Drosophila*[62], but also provide a promising strategy to safeguard against potential spread in prokaryotes via any possible horizontal gene transfer, especially when transgenes are intended for field releases.

## Methods
**Antibiotics and antibiotic-resistance genes**. Puromycin is a water soluble aminonucleoside antibiotic produced by the bacterium *Streptomyces alboniger*; it inhibits protein synthesis by disrupting peptide transfer on ribosomes, causing

premature chain termination during translation, and is thus a potent translational inhibitor in both prokaryotic and eukaryotic cells[63]. Geneticin (G418) is a water soluble aminoglycoside antibiotic produced by the bacterium *Micromonospora rhodorangea*; it interferes with 80S ribosomes and protein synthesis, and is therefore commonly used as a selective agent for eukaryotic cells (www.thermofisher.com). *PuroR* (*pac*)[38] from *Streptomyces alboniger* encodes *puromycin N-acetyltransferase*, and its expression in bacteria and mammalian cells confers resistance to puromycin. *NeoR* (*neo*)[37] encodes *aminoglycoside phosphotransferase*, and its expression in bacteria and mammalian cells confers resistance to geneticin, kanamycin, and Geneticin® (G418).

**Molecular biology**. To support transgenesis in diverse insect species, genetic constructs were built inside a PB JQ352761 plasmid[64] digested with FseI and AvrII. The PB (formerly IFP2) transposon was originally defined in the *Trichoplusia ni* cabbage looper moth[65] and has become a transposon of choice for genetically engineering of a wide variety of species, particularly insects[47]. We used Gibson assembly to engineer the genetic constructs. The protein sequences of *PuroR* (*pac*)[38] and *NeoR* (*neo*)[37] were back translated, codon optimized for *Drosophila* in Gene Designer 2.0 (https://www.dna20.com/resources/genedesigner), and synthesized as gene-blocks by Integrated DNA Technologies®. Both genes were expressed constitutively under *Opie2* regulatory sequences that originated from the baculovirus *Orgyia pseudotsugata* multicapsid nuclear polyhedrosis virus[42] and were amplified from the JQ352760 plasmid[64]. The transformation marker dsRed, a red fluorescent protein, was amplified from the KC733875 plasmid[66] and was ubiquitously expressed under the *Hr5IE1* regulatory sequence amplified from the KC991096 plasmid[67]. The SV40 3′UTR sequence from pAc5.1-V5-HisB (Invitrogen®) terminated the transcription of the transgenes. To confer the sex-specific translation of *PuroR* and *NeoR* genes, the *D. melanogaster tra* female-specific intron (*traF*) located between *tra* exon 1 and 2 (3L: 16591003-16590756, 248 bases, FlyBase.org) or the *D. melanogaster dsx* male-specific intron (*dsxM*) located between *dsx* exon 3 and 5 (3R: 7930688–7935767, 5080 bases) was inserted inside coding sequences of antibiotic-resistance genes. The plasmids generated in the study and their complete sequences (Fig. 1c) were deposited at www.addgene.org (#131613–#131618).

**Fly transgenesis**. The generated PB plasmids were injected into $w^{1118}$ flies at Rainbow Transgenic Flies, Inc. (http://www.rainbowgene.com). Recovered transgenic lines were balanced on the 2nd and 3rd chromosomes using single-chromosome balancer lines ($w^{1118}$; *CyO/sna$^{Sco}$* for II, and $w^{1118}$; *TM3, Sb$^1$/TM6B, Tb$^1$* for III) or a double-chromosome balancer line ($w^{1118}$; *CyO/Sp*; *Dr/TM6C, Sb$^1$*). Multiple independent lines integrated on the 2nd and/or 3rd chromosome were recovered for each plasmid and were tested on food supplemented with puromycin or geneticin. We used heterozygous transgenic lines harboring one copy of a transgene to assess the antibiotic resistance and sex-sorting efficiency. One transgenic line harboring the complete sex-sorter cassette integrated on the 3rd chromosome supported 100% sex sorting for both sexes, was homozygous fertile, and demonstrated especially good fitness. This line with two copies of the sex-sorter cassette (homozygous) was used for further analysis and was also deposited at the Bloomington Drosophila Stock Center (BDSC #79015).

**Genetics and sex selection**. Flies were maintained on cornmeal, molasses, and yeast medium (Old Bloomington Molasses Recipe) at 25 °C with a 12H/12H light/dark cycle. To assess drug resistance and/or sex selection, we used Instant *Drosophila* Food (Formula 4–24) from the Carolina Biological Supply Company. Per fly vial (FlyStaff.com), 1.1 g of dry food was mixed with 5 ml of distilled water supplemented with puromycin (Sigma #P8833) or geneticin (G418, Sigma #A1720) in varying concentrations from 0 to 1.2 mg/ml. To assess the drug resistance and/or sex sorting of the transgenic flies, a mixture of wt and transgenic flies harboring one copy of a transgene were allowed to breed on the supplemented food. Once the third instar larvae began to appear, the parents were removed. After hatching, the adult offspring, their transgenic markers, and their sex were recorded. Two or three independent transgenic lines integrated on the 2nd or 3rd chromosome were analyzed on food supplemented with puromycin and/or geneticin at 0.4 and 1.0 mg/ml. Since puromycin is known to be unstable over time in water solutions, we counted emerging flies and noted their sex for only 7 days after the first fly emerged. Three or five replicates were performed for each concentration.

**Fitness estimation**. To compare fly fitness on different food regimens, we calculated the percentage of embryos that survived to adulthood on the Instant Drosophila Food (Formula 4–24). Large numbers of *Drosophila* embryos were staged and collected on grape juice agar plates that were fitted into embryo collection cages (Genesee Scientific, FS59-100) following the stated protocol. In short, 50–80 flies were transferred into embryo collection cages and laid many eggs on grape juice plates fitted on the bottom of a cage. The grape juice plates were replaced, and the embryos that were laid on the plates overnight were collected in the morning. Embryos of both wt and transgenic flies harboring two copies of the sex-sorter cassette were collected. Batches of seventy five embryos were placed on the instant food supplemented with 0.0, 0.4, and 1.2 mg/ml of puromycin or geneticin, and the number and sex of the emerging adult flies were recorded for each condition. For the sex-sorter line raised on foods with different antibiotic

concentrations, the embryo-to-adult survival rate of either males or females on drug-supplemented foods was compared with that of the corresponding sex for wt embryos developed on the food without any antibiotics. In other words, the survival percentages for each sex were compared and presented as normalized percentages (Fig. 3d). The embryo-to-adult survival rate was estimated from five biological replicates. To assess the mating competence of the sex-sorted males, the males raised on the food supplemented with puromycin at the concertation of 1.2 ml/ml were placed into vials with virgin wt females and their progeny was scored for dsRed.

**Drug selection stage**. To determine the larval stage at which sex selection occurs, we fed first instar wt larvae with a yeast paste with or without the drug at 0.4 mg/ml and observed the larval stage to which they survived. *D. melanogaster* embryos were staged and collected on grape juice agar plates for a comparison of the fitness. Then, the embryos were transferred on agar plates whose surfaces were spread with a yeast paste supplemented with drugs. The plates were incubated at 25 °C, and the endpoint of embryo development was observed and recorded.

**Statistical analysis**. Statistical analysis was performed in JMP 8.0.2 by SAS Institute Inc. The percentages of a specific sex or embryo-to-adult survival were compared with the corresponding values estimated for the wt flies (Fig. 3a–c). P values were calculated for a two-sample Student's *t* test with equal variance.

**Reporting summary**. Further information on research design is available in the Nature Research Reporting Summary linked to this article.

## Data availability

All data underlying Figs. 1, 2, and 3 are represented fully within Supplementary Data 1–3. The plasmids constructed in the study (Fig. 1c) were deposited at Addgene.org (#131613 – #131618). The homozygous sex-sorter lines was deposited at Bloomington Drosophila Stock Center (#79015). The remaining *Drosophila* lines will be made available upon request. Any other relevant data are available from the authors upon reasonable request.

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

## Acknowledgements

We thank Pavel Nagorny for drawing the chemical structures of puromycin and geneticin. This work was supported in part by NIH grants 5K22AI113060, 1R21AI123937, and UCSD startup funds awarded to O.S.A., as well as NIH OD003878 awarded to B.A.H.

## Author contributions

O.S.A conceived the idea, engineered plasmids, and carried out preliminary experiments. O.S.A and N.P.K designed experiments. N.P.K, A.D.H., and J.L. performed all molecular and genetic experiments. All authors analyzed the data, contributed to the writing of the manuscript, and approved the final manuscript.

## Competing interests

O.S.A and B.A.H filed the US patent application (#20150237838) describing this technology. O.S.A has an equity interest in Agragene, Inc. and serves on the company's Scientific Advisory Board. All the other authors have no competing interests.
