## [Peer Review File · Nature Communications]

Reviewers' comments:

Reviewer #1 (Remarks to the Author):

Key results: The authors devise a novel sex-selection cassette in *Drosophila melanogaster* that permits isolation of a single sex when raised on antibiotics. The sex-selection method uses a transgene cassette including the antibiotic resistance genes NeoR and PuroR, which confer resistance to geneticin and puromycin, respectively. The authors engineered these antibiotic resistance genes to incorporate the introns TraF and DsxM, which are spliced in a sex-specific manner to produce a functional NeoR copy in females and a functional PuroR copy in males. This cassette is not detrimental to organismal fitness, and as sex-specific introns are common to a number of insect species, a similar approach can be applied to other organisms, making this sex-selection method a potentially important tool in control of pest insect populations.

Originality and significance: The authors highlight that this sex-selection method is of immense use in insect population control, which frequently relies on large numbers of sterile males. Although the narrative is presented from an angle of importance to insect control, it would increase the significance of the work if the value of this selection cassette to fly geneticists was also highlighted in the Introduction and Discussion. (Admirably, the authors have already deposited plasmids and stocks in the relevant repositories.)

Data and methodology: As mentioned in the below comments, some graphs could be altered to improve legibility and ease of data interpretation. Methodology is recorded in sufficient detail to reproduce results.

Appropriate use of statistics and treatment of uncertainties: The use of statistics in all data is appropriate.

Suggested improvements (all minor):

There are additional genetic methods of selection for females in *Drosophila*. Ashburner et al. (*Drosophila, A Laboratory Handbook*, pp. 135-6) describe Y-linked conditional or binary expression systems. McMahan et al. 2013 (doi 10.1534/g3.113.006411) describe a UAS::rpr transgene on the Y chromosome that can be used to male progeny in crosses with a GAL4 transgene.

Acronyms such as SIT, RIDL, TSL and IIT that are used relatively infrequently can be written out in their entirety to improve readability.

Insects have sexes, not genders.

[Numbers below refer to line numbers]

31: Change "geneticin supplements" to "geneticin-supplemented food" for clarity

41: In "Since then, SIT was successfully implemented..." change "was" to "has been"

61: Missing the word "transgene" after "conditional lethal"

94: Omit "the" before "first instar larvae"

169: Clarify that only males were recovered from all three of the transgenic lines

250: Although the system is resistant to loss-of-function mutations, over extended use it could select for gain-of-function mutants in which the tra or dsx introns can be spliced out in the undesired sex.

268: "the" should come before "functional" in "they harbor functional the sex-sorter gene cassette"

294: Missing "the" before "Tet-Off system"

305: The authors mention that they did not anticipate that the sex selection cassette would be detrimental to fly fitness, and instead they chose to assess the effect of different integration sites on fitness. However, they did assess fitness of homozygous copies of the cassette and found comparable fitness to wt flies, so this can be omitted.

In the Materials and Methods, fly genotypes are not italicized in several places.

382: Capitalize "biological" in "Carolina Biological Supply Company"

415: The wording "both males are equal at female mating" is confusing. Perhaps "both males have equal mating success?"

Figure 1:

608: "Insure" should be changed to "ensure" in "To insure that functional antibiotic-resistance proteins..."

Figure 2:

C and D could be changed to dot plots or box-and-whisker plots to improve interpretation of data.

Figure S1:

It would help in interpreting the data if A and B were displayed as dot plots or box-and-whisker plots rather than bar graphs. It would be easier to compare the sex ratios in C and D if they were displayed as stacked bar graphs as in Figure 2A and B.

B: Change the Y axis to % survival (Transgenic/wild-type)

C and D: Puromycin and geneticin sections should be moved so that they are on the same side in both C and D. "Gender percentage" as a Y axis title is a bit confusing and could be changed.

Table S3: "SexingLine" should be changed to "sex-sorter line" to be consistent with the text.

Reviewer #2 (Remarks to the Author):

This manuscript by Kandul et al. provide a sophisticated and intelligent design to cause sex separation in *Drosophila melanogaster* by the use of different antibiotics. While this is a clever approach, this is a strategy that should be strictly restricted to lab use!!!! To propose this approach for application in pest control is unreasonable and irresponsible!! Insecticide and antibiotic resistance are two major world wide problems, which should not be made even more severe by any means! The proposition to release antibiotic resistant insects is therefore absolutely irresponsible! First, the animals will die at some point and their DNA will enter the environment for bacteria to take this DNA up. Secondly, the antibiotic resistance will provide the insect carriers with a potential selectable advantage. Thus, rare survivors might have under certain circumstances an advantage and the transgenes could be selected and their presence increase in the population. While this is unlikely, this will still provide NGOs an argument against such approaches. So far transgenic SIT approaches use only neutral (fluorescent markers) or negative (sterility lethality) transgenes. And this is good so, as there will be no positive natural selection possible! Scientists in this field should be careful in what to promote and what NOT to promote! They should not offer any unnecessary arguments for opponents of such approaches.

Besides that I will review the manuscript in the following based on scientific standards, as it is a good piece of science and could be published as a *Drosophila* technique. Maybe then not in *nature Communications* but elsewhere.

Major revisions:

1. Sell it as *Drosophila* technique not as SIT approach.

2. Why is it, that having two copies reduces the amount of antibiotics to get sex separation (starting line 176). This is not straight forward. Having more should require more and not less. At least if it is leakiness of expression!!! Or is this a homozygous effect? More sex-specific splicing when locus homozygous? That could be tried by getting two copies that are at different loci. Thus

combining #2 and #3 in a transheterozygous condition!

3. The competition experiments are not competitive! They need to be carried out correctly or deleted. Giving two males to 10 virgin females is not a competitive situation. Giving 10 males of one kind and 10 males of the other kind to 10 virgins would be a competitive situation. Virgin *Drosophila* females, once old enough, will mate even with males that almost don't move and are not competitive at all. Thus either drop all sentences to competitiveness of the males or do the experiments correctly.

Minor edits:

Use of articles and singular/plural should be checked carefully throughout the manuscript (also Figure legends!!)

Line 28: *tra* and *dsx* are recessive lethal alleles and not capitalized. Please check this throughout the manuscript, as this is changing again and again. Also check the Figure legends.

Line 49: References 8 and 9 do not refer to IIT. They are referring to an approach originally called Eliminate Dengue (in Australia) and is now called "World Mosquito Program". However this is not IIT as the spread of *Wolbachia* is intended. This is more a gene drive scenario for population replacement! IIT is the approach by Stephen Dobson (Mosquito Mate). Thus publications by him should be cited in this respect!

Line 59: There is a sole focus on mosquitoes. What about references to Tephritid fruit flies (pupal colours) or Tse-Tse flies (hatch time) in respect to sexing methods!

Line 70: The first paper showing this type of sexing is not cited: Fu et al. *Nat Biotechnol.* 2007 Mar;25(3):353-7. Reference 30 is too old to be correct here! This one could be replaced by Mesa et al. 2018: Comparison of classical and transgenic genetic sexing strains of Mediterranean fruit fly (Diptera: Tephritidae) for application of the sterile insect technique. doi.org/10.1371/journal.pone.0208880.

Lines 95 to 97. While there is resistance to loss of function mutations, there could be mutations that interfere with sex-specific splicing and thereby disrupt the sexing. To present this as completely resistant to mutations is wrong and should not be done!

Line 115 and elsewhere (e.g. line 272): *piggyBac* (in italics) Not capitalized or any other way!

Lines 160 to 161: Splicing is not necessary affecting expression! Thus it is not the expression of genes, but "the production ofresistance can be limited..."

Lines 229- : The assay used is not conclusive for competition! See above! Thus either do a real competitive experiment or remove these statements!

Line 244: there is an "over " missing: advantages over

Line 252: Yes LOFs will be selected against, but mutations in the construct causing the loss of sex-specific splicing could generate a problem. Might be more rare indeed but should be mentioned!

Line 255: What are Y-autosome or X-autosomes? (probably chromosomes or translocations?)

Line 281: NO, the *Drosophila tra* intron is not conserved in other species. Actually in *Drosophila* female *tra* splicing is performed by *Sxl* and not by *TRA* itself, as in other species such as *C. capitata*. Thus if the authors had used the *C.c.* *tra* intron yes it could be transferable (as shown in the not cited Fu et al., 2007) but not the *D.m.* *tra*!!!

Lines 287 to 289: The promotion of using insecticide resistance genes in such approaches is ludicrous! Flooding the environment even more with things we don't want to have there! No No No !!!!

Lines 291 - : the positive selection also has its drawbacks, as it can be used as an argument that these insects have an advantage and could be selected in case rare SIT survivors carry these genes! Negative selection has here a great advantage as selling point for release!

Lines 311- : Drop sentences to mating competitiveness or do the experiments correctly!

Line 328 Point 7: No!!! Flooding the environment with antibiotic resistance genes is NOT sellable! PERIOD!

Line 362: Where exactly were the introns integrated. Any thought or specific consideration on the exact sequence position as made in the non-cited Fu et al., 2007? Sequences outside the intron might influence the efficiency of sex-specific splicing!

Line 370: "Sp" is a dominant mutation!

Lines 410- : One and one male for 10 females is not a competitive situation!! This is not a competition experiment!

Line 439: Only one line was deposited

Figure 1 legend. Text under (b) is referring still to (a). There is no (b) text, which could be: survival of females and/or males carrying the respective constructs on antibiotic selection (pur, blue) (gen, green)

Figure S1: This should become a main Figure. This provides the clear data!
Panel b) transgenic / wild type. This cannot be true. This cannot be a ratio! A ratio compared to 0 does not work. It is probably survival percentage or something similar!

Line 652: 1 mg/mL NOT 0.1!!!!

**Reviewers' comments:****Reviewer #1 (Remarks to the Author):**

Key results: The authors devise a novel sex-selection cassette in *Drosophila melanogaster* that permits isolation of a single sex when raised on antibiotics. The sex-selection method uses a transgene cassette including the antibiotic resistance genes NeoR and PuroR, which confer resistance to geneticin and puromycin, respectively. The authors engineered these antibiotic resistance genes to incorporate the introns TraF and DsxM, which are spliced in a sex-specific manner to produce a functional NeoR copy in females and a functional PuroR copy in males. This cassette is not detrimental to organismal fitness, and as sex-specific introns are common to a number of insect species, a similar approach can be applied to other organisms, making this sex-selection method a potentially important tool in control of pest insect populations.

Originality and significance: The authors highlight that this sex-selection method is of immense use in insect population control, which frequently relies on large numbers of sterile males. Although the narrative is presented from an angle of importance to insect control, it would increase the significance of the work if the value of this selection cassette to fly geneticists was also highlighted in the Introduction and Discussion. (Admirably, the authors have already deposited plasmids and stocks in the relevant repositories.)

Data and methodology: As mentioned in the below comments, some graphs could be altered to improve legibility and ease of data interpretation. Methodology is recorded in sufficient detail to reproduce results.

Appropriate use of statistics and treatment of uncertainties: The use of statistics in all data is appropriate.

Suggested improvements (all minor):

There are additional genetic methods of selection for females in *Drosophila*. Ashburner et al. (*Drosophila, A Laboratory Handbook*, pp. 135-6) describe Y-linked conditional or binary expression systems. McMahan et al. 2013 (doi 10.1534/g3.113.006411) describe a UAS::rpr transgene on the Y chromosome that can be used to male progeny in crosses with a GAL4 transgene.

Thank you, we corrected the current text accordingly.

Acronyms such as SIT, RIDL, TSL and IIT that are used relatively infrequently can be written out in their entirety to improve readability.

Thank you for the detailed look at our manuscript. Infrequently used acronyms are now spelled out throughout the manuscript. At the same time, SIT, RIDL, and IIT are the major methods for insect control and as such are frequently referred only by their acronyms in the press.

Insects have sexes, not genders.

This has been changed throughout the manuscript.

[Numbers below refer to line numbers]

31: Change “geneticin supplements” to “geneticin-supplemented food” for clarity

This has been corrected.

41: In “Since then, SIT was successfully implemented...” change “was” to “has been”

This is no longer applicable with the significant text changes.

61: Missing the word “transgene” after “conditional lethal”

This has been corrected.

94: Omit “the” before “first instar larvae”

This has been corrected.

169: Clarify that only males were recovered from all three of the transgenic lines

This has been corrected.

250: Although the system is resistant to loss-of-function mutations, over extended use it could select for gain-of-function mutants in which the tra or dsx introns can be spliced out in the undesired sex.

This has been corrected, now we refer to loss-of-function mutations in the text. Gain-of-function mutations indeed can happen but they are much less frequent than loss-of-function mutations.

268: “the” should come before “functional” in “they harbor functional the sex-sorter gene cassette”

This has been corrected.

294: Missing “the” before “Tet-Off system”

Sentence has been rearranged and is now correct.

305: The authors mention that they did not anticipate that the sex selection cassette would be detrimental to fly fitness, and instead they chose to assess the effect of different integration sites on fitness. However, they did assess fitness of homozygous copies of the cassette and found comparable fitness to wt flies, so this can be omitted.

This has been deleted.

In the Materials and Methods, fly genotypes are not italicized in several places.

This has been corrected.

382: Capitalize “biological” in “Carolina Biological Supply Company”

This has been corrected.

415: The wording “both males are equal at female mating” is confusing. Perhaps “both males have equal mating success?”

This has been corrected.

Figure 1:

608: “Insure” should be changed to “ensure” in “To insure that functional antibiotic-resistance proteins...”

This has been corrected.

Figure 2:

C and D could be changed to dot plots or box-and-whisker plots to improve interpretation of data.

We added the data points to the bar graphs to improve presentation of the data.

Figure S1:

It would help in interpreting the data if A and B were displayed as dot plots or box-and-whisker plots rather than bar graphs. It would be easier to compare the sex ratios in C and D if they were displayed as stacked bar graphs as in Figure 2A and B.

B: Change the Y axis to % survival (Transgenic/wild-type)

C and D: Puromycin and geneticin sections should be moved so that they are on the same side in both C and D. “Gender percentage” as a Y axis title is a bit confusing and could be changed.

We updated the figures following your comments. Thank you.

Table S3: “SexingLine” should be changed to “sex-sorter line” to be consistent with the text.

This has been corrected.

Reviewer #2 (Remarks to the Author):

This manuscript by Kandul et al. provide a sophisticated and intelligent design to cause sex separation in *Drosophila melanogaster* by the use of different antibiotics. While this is a clever approach, this is a strategy that should be strictly restricted to lab use!!!! To propose this approach for application in pest control is unreasonable and irresponsible!! Insecticide and

antibiotica resistance are two major world wide problems, which should not be made even more severe by any means! The proposition to release antibiotica resistant insects is therefore absolutely irresponsible! First, the animals will die at some point and their DNA will enter the environment for bacteria to take this DNA up. Secondly, the antibiotica resistance will provide the insect carriers with a potential selectable advantage. Thus, rare survivors might have under certain circumstances an advantage and the transgenes could be selected and their presence increase in the population. While this is unlikely, this will still provide NGOs an argument against such approaches. So far transgenic SIT approaches use only neutral (fluorescent markers) or negative (sterility lethality) transgenes. And this is good so, as there will be no positive natural selection possible! Scientists in this field should be careful in what to promote and what NOT to promote! They should not offer any unnecessary arguments for opponents of such approaches.

Besides that I will review the manuscript in the following based on scientific standards, as it is a good piece of science and could be published as a *Drosophila* technique. Maybe then not in *nature Communications* but elsewhere.

Major revisions:

1. Sell it as *Drosophila* technique not as SIT approach.

We understand your concerns raised by our work, and agree that as originally presented, this technology could have unexpected and dangerous consequences. We have therefore adjusted the paper accordingly. Please see the altered focus in the abstract, introduction, and discussion, which now also includes a caution against using this technology in the field. We also suggest that a simple strategy to safeguard against the potential horizontal transfer and spread of transgene in prokaryotes. The insertion of a regular intron to break the coding sequence of the transgene that is intended for the field releases would block it splicing and function in any prokaryote. We successfully used this strategy to clone and manipulate multiple genes that are otherwise 'toxic' for *E. coli*, such as restriction endonucleases and T4 DNA ligase, etc. (<https://www.nature.com/articles/ncomms13100>), in the lab..

2. Why is it, that having two copies reduces the amount of antibiotics to get sex separation (starting line 176). This is not straight forward. Having more should require more and not less. At least if it is leakiness of expression!!! Or is this a homozygous effect? More sex-specific splicing when locus homozygous? That could be tried by getting two copies that are at different loci. Thus combining #2 and #3 in a transheterozygous condition!

We think this is the homozygous effect that improves sex-specific splicing. Since both expression level and sex specific splicing depend to the genome location, we analysed multiple insertions of every tested transgene to identify the transgenic line, in which each transgene operated properly. We found that a single copy of *PuroR^{dsxM}* or *NeoR^{traF}* were spliced in both sexes at the tested insertion sites with an antibiotic concentration of 0.4mg/mL; but *PuroR^{dsxM}* or *NeoR^{traF}* was spliced more efficiently in males or females, respectively, at one established

insertion line, and this became apparent at the higher antibiotic concentration. We were able to homozygous one of two 'working' (at 1.0mg/mL) insertion lines of the sex-sorter cassette (*PuroR^{dsxM}* + *NeoR^{traF}*) and found that two-copies of the cassette operated properly at the lower concentration (0.4mg/mL) than one copy did (1.0mg/mL). The sex-specific splicing of sex-determination genes are known to be self-reinforcing process, therefore two copies of the sex-sorter cassette located at the homologous location could reinforce the correct splicing and suppress the incorrect splicing by recruiting SXL, TRA, and other proteins to their genomic locations. We agree that the combining the lines #2 and #3 of the sex-sorter cassette in transheterozygous conditions would be an interesting test to tease apart the homologous location effect. Unfortunately, it goes beyond the scope of this paper.

3. The competition experiments are not competitive! They need to be carried out correctly or deleted. Giving two males to 10 virgin females is not a competitive situation. Giving 10 males of one kind and 10 males of the other kind to 10 virgins would be a competitive situation. Virgin *Drosophila* females, once old enough, will mate even with males that almost don't move and are not competitive at all. Thus either drop all sentences to competitiveness of the males or do the experiments correctly.

We removed the male competition assay.

Minor edits:

Use of articles and singular/plural should be checked carefully throughout the manuscript (also Figure legends!!)

We thank you for pointing this out, and have subsequently had the entire manuscript professionally edited for language.

Line 28: *tra* and *dsx* are recessive lethal alleles and not capitalized. Please check this throughout the manuscript, as this is changing again and again. Also check the Figure legends.

This has been corrected.

Line 49: References 8 and 9 do not refer to IIT. They are referring to an approach originally called Eliminate Dengue (in Australia) and is now called "World Mosquito Program". However this is not IIT as the spread of *Wolbachia* is intended. This is more a gene drive scenario for population replacement! IIT is the approach by Stephen Dobson (Mosquito Mate). Thus publications by him should be cited in this respect!

Thank you. This has been corrected.

Line 59: There is a sole focus on mosquitoes. What about references to Tephritid fruit flies (pupal colours) or Tse-Tse flies (hatch time) in respect to sexing methods!

We added more comments to the sexing methods used for Tephritid fruit flies (Medfly and Mexfly) and Tsetse flies, and toned down discussion of mosquito sexing methods especially in the introduction. The Medfly sexing method (the linkage of the *white pupae + temperature sensitive lethal* loci) is the most advanced sexing methods and is used for high-throughput production of Medfly males by culling females. Still, it was established serendipitously with classic genetic methods and could not be transferred to other Tephritid fruit fly species.

Line 70: The first paper showing this type of sexing is not cited: Fu et al. Nat Biotechnol. 2007 Mar;25(3):353-7. Reference 30 is too old to be correct here! This one could be replaced by Mesa et al. 2018: Comparison of classical and transgenic genetic sexing strains of Mediterranean fruit fly (Diptera: Tephritidae) for application of the sterile insect technique. doi.org/10.1371/journal.pone.0208880.

Thank you, we added both references, and removed ref. #30.

Lines 95 to 97. While there is resistance to loss of function mutations, there could be mutations that interfere with sex-specific splicing and thereby disrupt the sexing. To present this as completely resistant to mutations is wrong and should not be done!

This has been changed to be more specific. The described system is resistant to loss-of-function mutations, gain-of-function mutations can still disrupt sex-sorting, though gain-of-function mutations are much less frequent than loss-of-function ones.

Line 115 and elsewhere (e.g. line 272): *piggyBac* (in italics) Not capitalized or any other way!

This has been corrected.

Lines 160 to 161: Splicing is not necessary affecting expression! Thus it is not the expression of genes, but "the production ofresistance can be limited..."

This has been corrected.

Lines 229- : The assay used is not conclusive for competition! See above! Thus either do a real competitive experiment or remove these statements!

We removed the male competition assay.

Line 244: there is an "over " missing: advantages over

This has been corrected.

Line 252: Yes LOFs will be selected against, but mutations in the construct causing the loss of sex-specific splicing could generate a problem. Might be more rare indeed but should be mentioned!

This has been corrected.

Line 255: What are Y-autosome or X-autosomes? (probably chromosomes or translocations?)

Translocation. We removed this unclear writing.

Line 281: NO, the Drosophila tra intron is not conserved in other species. Actually in Drosophila female tra splicing is performed by Sxl and not by TRA itself, as in other species such as C. capitata. Thus if the authors had used the C.c. tra intron yes it could be transferable (as shown in the not cited Fu et al., 2007) but not the D.m. tra!!!

We completely agree with the raised point and updated the text accordingly.

Lines 287 to 289: The promotion of using insecticide resistance genes in such approaches is ludicrous! Flooding the environment even more with things we don't want to have there! No No No !!!!

Please see our response above.

Lines 291 - : the positive selection also has its drawbacks, as it can be used as an argument that these insects have an advantage and could be selected in case rare SIT survivors carry these genes! Negative selection has here a great advantage as selling point for release!

We agree that the advantage and the selection depends on the context, and we hope that antibiotic concentrations in the wild are not that high. That said, of course these things should be denoted and implemented in an environment specific manner. If the antibiotic concentrations are used in areas for cattle or plant protection (which they sometimes are) this may benefit or not the strategy, depending on the goal whether it is useful to have the engineered males have a fitness advantage.

Lines 311- : Drop sentences to mating competitiveness or do the experiments correctly!

We excluded the male competition assay..

Line 328 Point 7: No!!! Flooding the environment with antibiotic resistance genes is NOT sellable! PERIOD!

We agree and corrected the text accordingly (please see our response above)

Line 362: Where exactly were the introns integrated. Any thought or specific consideration on the exact sequence position as made in the non-cited Fu et al., 2007? Sequences outside the intron might influence the efficiency of sex-specific splicing!

Both introns were integrated right after the start codon (ATG) of antibiotic-resistance genes. We submitted the sequences and map of every construct to AddGene.org, after publication these plasmid and sequence will be released. We used the consensus sequences for Drosophila splice junctions, to match the most frequent bases adjacent to an intron sequence (..TG[gt---ag]AT/C..).

Line 370: "Sp" is a dominant mutation!

This has been corrected.

Lines 410- : One and one male for 10 females is not a competitive situation!! This is not a competition experiment!

We do not use the word “competitiveness” in the current text.

Line 439: Only one line was deposited.

We deposited one homozygous lines with the complete sex-sorter cassette, since it contains all genetic parts and can be used to generate males or females. Bloomington Drosophila Stock Center prefers to keep Drosophila lines that are requested by multiple researchers and ‘retires’ unused stocks.

Figure 1 legend. Text under (b) is referring still to (a). There is no (b) text, which could be: survival of females and/or males carrying the respective constructs on antibiotic selection (pur, blue) (gen, green)

This has been corrected.

Figure S1: This should become a main Figure. This provides the clear data!

We added the data from this figure to the current figures 1 and 2.

Panel b) transgenic / wild type. This cannot be true. This cannot be a ratio! A ratio compared to 0 does not work. It is probably survival percentage or something similar!

This is the survival percentage. The heterozygous not-yet-balanced transgenic flies and *w^t* flies were raised on the food supplemented with puromycin or geneticin.

Line 652: 1 mg/mL NOT 0.1!!!!

This has been corrected.